# (D-Ala^2^)GIP Inhibits Inflammatory Bone Resorption by Suppressing TNF-α and RANKL Expression and Directly Impeding Osteoclast Formation

**DOI:** 10.3390/ijms25052555

**Published:** 2024-02-22

**Authors:** Angyi Lin, Hideki Kitaura, Fumitoshi Ohori, Takahiro Noguchi, Aseel Marahleh, Jinghan Ma, Jiayi Ren, Mariko Miura, Ziqiu Fan, Kohei Narita, Itaru Mizoguchi

**Affiliations:** 1Division of Orthodontics and Dentofacial Orthopedics, Tohoku University Graduate School of Dentistry, 4-1 Seiryo-Machi, Aoba-Ku, Sendai 980-8575, Miyagi, Japan; lin.angyi.r5@dc.tohoku.ac.jp (A.L.); fumitoshi.ohori.t3@dc.tohoku.ac.jp (F.O.); takahiro.noguchi.r4@dc.tohoku.ac.jp (T.N.); ma.jinghan.s1@dc.tohoku.ac.jp (J.M.); ren.jiayi.p7@dc.tohoku.ac.jp (J.R.); mariko.miura.b5@tohoku.ac.jp (M.M.); fan.ziqiu.q1@dc.tohoku.ac.jp (Z.F.); kohei.narita.a2@tohoku.ac.jp (K.N.); mizo@tohoku.ac.jp (I.M.); 2Frontier Research Institute for Interdisciplinary Sciences, Tohoku University, Sendai 980-8575, Miyagi, Japan; marahleh@tohoku.ac.jp

**Keywords:** GIP, osteoclast, bone resorption, inflammation, LPS, TNF-α, RANKL, osteoblast, macrophage

## Abstract

Glucose-insulinotropic polypeptide (GIP) is an incretin hormone that induces insulin secretion and decreases blood glucose levels. In addition, it has been reported to suppress osteoclast formation. Native GIP is rapidly degraded by dipeptidyl peptidase-4 (DPP-4). (D-Ala^2^)GIP is a newly developed GIP analog that demonstrates enhanced resistance to DPP-4. This study aimed to evaluate the influence of (D-Ala^2^)GIP on osteoclast formation and bone resorption during lipopolysaccharide (LPS)-induced inflammation in vivo and in vitro. In vivo, mice received supracalvarial injections of LPS with or without (D-Ala^2^)GIP for 5 days. Osteoclast formation and bone resorption were evaluated, and TNF-α and RANKL expression were measured. In vitro, the influence of (D-Ala^2^)GIP on RANKL- and TNF-α-induced osteoclastogenesis, LPS-triggered TNF-α expression in macrophages, and RANKL expression in osteoblasts were examined. Compared to the LPS-only group, calvariae co-administered LPS and (D-Ala^2^)GIP led to less osteoclast formation, lower bone resorption, and decreased TNF-α and RANKL expression. (D-Ala^2^)GIP inhibited osteoclastogenesis induced by RANKL and TNF-α and downregulated TNF-α expression in macrophages and RANKL expression in osteoblasts in vitro. Furthermore, (D-Ala^2^)GIP suppressed the MAPK signaling pathway. The results suggest that (D-Ala^2^)GIP dampened LPS-triggered osteoclast formation and bone resorption in vivo by reducing TNF-α and RANKL expression and directly inhibiting osteoclastogenesis.

## 1. Introduction

The incidence of diabetes mellitus has increased worldwide in recent decades. The number of patients globally is projected to reach 783.2 million over the next 20 years [1]. Diabetic complications include cardiovascular diseases, liver diseases, cognitive disabilities, and cancer [2,3,4,5,6,7]. Bone fracture risk is also elevated in diabetic patients compared to their healthy counterparts owing to reduced bone strength [8].

Glucose-dependent insulinotropic polypeptide (GIP), also known as gastric inhibitory polypeptide, is an incretin hormone secreted by intestinal K cells [9]. By attaching to its receptor (GIPr) on pancreatic β-cells, it promotes insulin secretion, thereby decreasing blood glucose levels [10,11]. Native GIP is susceptible to rapid degradation by dipeptidyl peptidase-4 (DPP-4), an enzyme that is present in blood with a half-life of 5–7 min [12,13]. Several N-terminally modified and long-acting GIP analogs have been developed to enhance the resistance to DPP-4 and increase the duration of the stimulating effect on the GIPr. (D-Ala^2^)GIP, the reagent used in the present study, is an analog with a d-alanine at position 2 [14].

The GIPr is also widely expressed in many tissues outside the pancreas, including muscle, adipose tissue, brain, and bone tissues [15]. Specifically, in bone tissue, the GIPr is expressed in osteoblasts, osteocytes, and osteoclasts [16,17,18]. Osteoclasts, which are myeloid-derived multinucleated cells, resorb bone tissue and play a vital role in bone remodeling [19]. Excessive bone resorption caused by osteoclasts under pathological conditions leads to various diseases including rheumatoid arthritis and periodontitis [20,21]. Macrophage colony-stimulating factor (M-CSF) and receptor activator of NF-kB ligand (RANKL) play crucial roles in promoting osteoclast formation and activation [22,23]. Moreover, tumor necrosis factor-α (TNF-α) is able to directly trigger osteoclastogenesis in vitro and in vivo [24,25,26,27].

Previous studies have indicated the potential role of GIP and its analogs in the downregulation of osteoclast formation and bone resorption under various conditions. In vivo, GIP alleviated bone resorption in healthy individuals as well as patients with Type 1 and 2 diabetes [28,29,30]. Intravenous injection of GIP recovered bone mineral density loss in ovariectomized rats [31]. Furthermore, subcutaneous injection of N-AcGIP, another N-terminal-modified DPP-4 resistant long-acting GIP analog, decreased osteoclast-mediated bone resorption in ovariectomized mice [32]. In vitro, (D-Ala^2^)GIP suppresses RANKL-induced osteoclast formation in a dose-dependent manner by reducing Ca^2+^ influx triggered by RANKL stimulation [32]. Moreover, GIP inhibits parathyroid hormone (PTH)–induced bone resorption in a dose-dependent manner [18].

Several studies have evaluated the effects of GIP and its analogs on inflammation in various tissues. Contradictory results have been reported regarding adipose tissue. GIP treatment enhanced the expression of proinflammatory genes and macrophage infiltration in the adipose tissue of obese mice [33], while another study using (D-Ala^2^)GIP reported conversely [34]. GIP overexpression also reduced macrophage infiltration of atherosclerotic plaques in ApoE^−/−^ mice [35]. And aggravated inflammation in gingiva tissue appeared in a GIP receptor knockout (GIPRKO) periodontitis mouse model compared to WT mice with periodontitis [36].

Lipopolysaccharide (LPS) is an essential constituent of the outer membrane of Gram-negative bacteria that has been found to trigger inflammatory bone loss by robustly upregulating osteoclast formation and activation, leading to bone resorption [37,38,39]. It induces the expression of proinflammatory cytokines such as IL-1 and TNF-α at the inflammation site, which are involved in LPS-driven osteoclastogenesis and osteolysis [40,41,42]. LPS directly stimulates RANKL expression in osteoblasts and enhances TNF-α expression in macrophages, which can further increase RANKL production, thus synergistically aggravating osteoclast formation in inflammatory conditions [38,40,43].

To the best of our knowledge, there are currently no reports on the impact of GIP and its analogs on bone tissue under inflammatory conditions, particularly inflammation-triggered osteoclast formation and bone resorption. Therefore, the objective of the present study was to evaluate the effect of (D-Ala^2^)GIP, a DPP-4 resistant GIP analog, on osteoclast formation and bone resorption in LPS-induced inflammatory conditions in vivo and to explore its possible mechanism in in vitro experiments.

## 2. Results

### 2.1. (D-Ala^2^)GIP Inhibited LPS-Induced Bone Resorption In Vivo

The ratio of the bone resorption area to the entire calvaria was assessed after micro-computed tomography (CT) scanning. A remarkably larger bone resorption area was observed in the LPS-only group compared with groups received PBS or (D-Ala^2^)GIP. Mice co-administered LPS and (D-Ala^2^)GIP demonstrated decreased bone resorption compared with those treated with LPS alone (Figure 1A,B).

### 2.2. (D-Ala^2^)GIP Inhibited LPS-Induced Osteoclast Formation In Vivo

To investigate the impact of (D-Ala^2^)GIP on LPS-triggered osteoclastogenesis in vivo, LPS alone or with (D-Ala^2^)GIP was injected subcutaneously into mouse calvariae for 5 consecutive days. The LPS-only group displayed a larger number of multinucleated TRAP-positive osteoclasts in the suture mesenchyme than groups receiving PBS or (D-Ala^2^)GIP. Significantly fewer osteoclasts were observed in mice co-administered LPS and (D-Ala^2^)GIP than in those that received LPS alone (Figure 2A,B). Moreover, the LPS-only group showed the highest mRNA expression levels of TRAP and cathepsin K, whereas lower levels were observed in mice injected with both LPS and (D-Ala^2^)GIP (Figure 2C,D).

### 2.3. (D-Ala^2^)GIP Suppressed Expression of TNF-α and RANKL In Vivo

The TNF-α and RANKL mRNA expression levels of calvariae bone pieces were evaluated. The LPS-only group showed enhanced expression of the two cytokines compared to the PBS and (D-Ala^2^)GIP groups. The co-administered LPS and (D-Ala^2^)GIP group displayed a decreased level of TNF-α and RANKL mRNA expression in comparison with the LPS-only group (Figure 3A,B).

### 2.4. (D-Ala^2^)GIP Decreased Osteoclastogenesis Triggered by RANKL and TNF-α and Had No Effect on Cell Viability of Osteoclast Precursors In Vitro

M-CSF and RANKL or TNF-α was added to osteoclast precursors to induce osteoclastogenesis, and the effects of (D-Ala^2^)GIP on osteoclast precursors were investigated. Many TRAP staining–positive multinucleated cells were observed in wells with M-CSF and RANKL or TNF-α. Fewer were observed when (D-Ala^2^)GIP was added (Figure 4A,B). To examine the effect of (D-Ala^2^)GIP on osteoclast precursor viability, the cells were treated with different concentrations of (D-Ala^2^)GIP (0, 2, 20, and 200 nM). No significant difference in cell viability was observed among the groups after culturing for 5 days (Figure 4C).

### 2.5. (D-Ala^2^)GIP Downregulated LPS-Driven TNF-α Expression in Macrophages and RANKL Expression in Osteoblasts In Vitro

Real-time reverse transcription polymerase chain reaction (RT-PCR) was conducted to analyze the TNF-α expression level of peritoneal macrophages. Elevated TNF-α expression appeared in the LPS-administered group compared with the PBS group and (D-Ala^2^)GIP group. Conversely, in the LPS and (D-Ala^2^)GIP group, reduced TNF-α expression compared to the LPS-only group was observed (Figure 5A). We also evaluated RANKL expression in osteoblasts by using RT-PCR. Compared to the PBS and (D-Ala^2^)GIP groups, RANKL expression was higher in the LPS group. However, in the LPS and (D-Ala^2^)GIP group, RANKL expression decreased compared to the LPS-only group (Figure 5B).

### 2.6. (D-Ala^2^)GIP Dampened LPS-Triggered Mitogen-Activated Protein Kinases (MAPKs) Signaling Pathway in Peritoneal Macrophages In Vitro

LPS or LPS with (D-Ala^2^)GIP was added to the culture dishes at the indicated time points (5, 15, 30 min); a control dish with no added material was marked 0 min. LPS treatment increased the levels of phosphorylated p38, ERK, and JNK, which peaked at 15 min. For p38 and JNK, (D-Ala^2^)GIP-treated cells showed a temporarily higher increase in phosphorylation at 5 min than in cells treated with LPS only. The significance disappeared at 15 min and was counteracted at 30 min for p38, whereas for JNK, the significance was reversed at 15 min. LPS-induced phosphorylation of ERK was inhibited by (D-Ala^2^)GIP after 15 min (Figure 6A–D).

### 2.7. (D-Ala^2^)GIP Dampened LPS-Triggered MAPKs Signaling Pathway in Osteoblasts In Vitro

Similarly, osteoblasts were treated with LPS or LPS plus (D-Ala^2^)GIP at the indicated time points (0, 5, 15, 30 min), and no reagent was added to the 0 min dishes. LPS increased the levels of phosphorylated p38, ERK, and JNK proteins, peaking at 30 min for p38 and JNK and at 5 min for ERK. (D-Ala^2^)GIP suppressed LPS-triggered phosphorylation of JNK at 5 min and inhibited ERK phosphorylation at 5, 15, and 30 min. It had no effect on p38 phosphorylation at any time points in the experiment (Figure 7A–D).

## 3. Discussion

The purpose of the present study was to evaluate the effect of (D-Ala^2^)GIP on LPS-induced osteoclast formation and bone resorption in vivo and explore its possible mechanisms via in vitro experiments. The results show that (D-Ala^2^)GIP significantly suppressed LPS-triggered osteoclastogenesis and bone resorption by modulating the expression levels of RANKL and TNF-α in vivo. In vitro, (D-Ala^2^)GIP directly impeded osteoclastogenesis induced by RANKL and TNF-α without affecting the cell viability of osteoclast precursors. Moreover, (D-Ala^2^)GIP dampened LPS-triggered TNF-α expression in macrophages as well as LPS-triggered RANKL expression in osteoblasts (Figure 8).

As the number of patients with diabetes is increasing, scientists have increasingly focused on developing novel diabetes therapies [1]. GIP, an incretin hormone, has attracted research interest owing to its role in promoting postprandial insulin secretion from β-cells in the pancreas [9,10,11]. However, native GIP can be rapidly cleaved into its inactive form by DPP-4 [12,13]. (D-Ala^2^)GIP, the long-acting DPP-4 resistant GIP analogue evaluated in this study, exerts an agonistic effect on the GIPr and has a longer half-life than endogenous GIP [14]. In recent decades, GIP has been identified as a part of the enteroendocrine–osseous axis in bone metabolism [44]. Contradictory results on bone histomorphometric alterations in GIPRKO mice have been reported, possibly due to differences in the deleted exons. Mice with exon 1–6 deletions displayed enhanced trabecular bone volume and decreased osteoclast numbers [45], while mice with exon 4–5 deletions exhibited decreased bone volume, fewer bone formation markers, and increased osteoclastic bone resorption [11,46]. Conversely, the administration of exogenous GIP and its analogs prevented the loss of bone mineral density and reduced bone resorption in ovariectomized rodent models [31,32], and dampened bone resorption in both healthy individuals and diabetic patients [28,29,30]. These studies suggested a potential function of GIP in inhibiting osteoclast formation and osteoclastic bone-resorbing activity in vivo. In the present study, we evaluated the effect of (D-Ala^2^)GIP on LPS-triggered osteoclastogenesis in mouse calvaria. Bone sections were subjected to histological analysis after 5 consecutive days of drug administration (LPS alone or LPS with (D-Ala^2^)GIP). Fewer osteoclasts were identified in the co-administered LPS and (D-Ala^2^)GIP group, suggesting that (D-Ala^2^)GIP ameliorated LPS-induced osteoclastogenesis. The mRNA expression levels of two essential osteoclastic markers, TRAP and cathepsin K, were evaluated, revealing lower expression in the co-administered LPS and (D-Ala^2^)GIP group compared to the LPS-only group. We also analyzed the bone destruction area using micro-CT, which was determined as the ratio of the bone resorption area to the whole calvaria. The results revealed a smaller bone resorption area in the group co-administered LPS and (D-Ala^2^)GIP than in the group that received only LPS. Our findings indicated that (D-Ala^2^)GIP alleviated LPS-triggered osteoclastogenesis and bone destruction in vivo and are in agreement with previous studies [28,29,30,31,32]. Conflicting findings have been reported regarding the effect of GIP on inflammation in adipose tissue. In one study, GIP treatment enhanced the expression of proinflammatory genes and macrophage infiltration in obese mice [33]. Conversely, another study using (D-Ala^2^)GIP reported the opposite [34]. The discrepancies might be due to the variances in factors including mouse models and dosage applied. In the present study, we observed an anti-inflammatory effect of (D-Ala^2^)GIP in bone tissue in WT mice, consistent with the latter study. The actions of GIP towards inflammation might also vary across different tissues.

Next, we explored the possible mechanisms underlying the inhibition of osteoclastogenesis and bone resorption in vivo. Two potential mechanisms were identified. First, we evaluated whether (D-Ala^2^)GIP suppressed LPS-induced expression of TNF-α and RANKL, the two proinflammatory cytokines that are known to foster osteoclast formation. Our results show that (D-Ala^2^)GIP downregulated TNF-α and RANKL expression in vivo, which suggests that (D-Ala^2^)GIP acted by suppressing the production of proinflammatory cytokines, thus decreasing osteoclast formation. We also examined whether (D-Ala^2^)GIP could directly reduce osteoclast formation. In our in vitro experiments, (D-Ala^2^)GIP inhibited osteoclastogenesis triggered by RANKL and TNF-α and had no impact on osteoclast precursor viability, which revealed that (D-Ala^2^)GIP directly impeded osteoclast formation.

We then conducted in-depth in vitro experiments to analyze the mechanisms underlying the inhibition of TNF-α and RANKL expression by (D-Ala^2^)GIP in vivo. Our results demonstrate that (D-Ala^2^)GIP mitigated TNF-α mRNA expression in macrophages, which enhanced RANKL expression and aggravated osteoclast formation synergistically. Previous studies have reported that LPS can induce the phosphorylation of p38, ERK, and JNK in macrophages, promoting cellular TNF-α expression [47,48]. In the present study, Western blot analysis revealed that (D-Ala^2^)GIP had an inhibitory effect on the LPS-induced phosphorylation of p38, ERK, and JNK in macrophages. However, for p38 and JNK, (D-Ala^2^)GIP-treated cells experienced a temporary increase in phosphorylation compared to those treated with LPS alone, and this effect was countered shortly thereafter. We suppose this suppressive effect, together with the alleviated ERK signaling, contributed to the inhibition of TNF-α expression in macrophages. The short-term increase in p38 and JNK signaling in (D-Ala^2^)GIP-treated cells requires further investigation. Our results indicate that (D-Ala^2^)GIP suppressed RANKL expression in osteoblasts. It has also been reported that LPS-induced phosphorylation of p38, ERK, and JNK led to upregulation of RANKL expression in osteoblasts [49,50]. In the present study, (D-Ala^2^)GIP relieved LPS-induced phosphorylation of ERK and JNK in osteoblasts, revealing that (D-Ala^2^)GIP dampened LPS-induced MAPKs signaling, resulting in inhibited RANKL expression in osteoblasts. Subsequently, the expression level of the two proinflammatory cytokines, TNF-α and RANKL was reduced in vivo and osteoclast formation as well as osteoclastic bone resorption was suppressed.

Our study demonstrated the inhibitory effect of (D-Ala^2^)GIP on osteoclast formation and bone resorption in an LPS-triggered acute inflammatory bone environment. (D-Ala^2^)GIP might also ameliorate bone destruction in chronic inflammation if applied long term, supporting advancements in therapies for inflammatory bone diseases. Further research is required for confirmation. In addition, the expression levels of proinflammatory cytokines including TNF-α and RANKL are also upregulated in diabetes mellitus, causing more bone loss [51,52,53]; therefore, (D-Ala^2^)GIP may serve as an alternative treatment for osteoporosis in diabetic patients, as well as those with inflammatory bone diseases accompanying diabetes.

## 4. Materials and Methods

### 4.1. Animal Model and Reagents

8–10-week-old male C57BL6/J mice were purchased from CLEA Japan (Tokyo, Japan) and maintained at the Animal Facility of Tohoku University. Four mice were randomly assigned to each experimental group and received reagent injections. All animal care and experimental procedures followed the guidelines of the Tohoku University Science Animal Care and Use Committee. *Escherichia coli* LPS was obtained from Sigma-Aldrich (St. Louis, MO, USA). (D-Ala^2^)GIP was purchased from Bachem (Bubendorf, Switzerland). Recombinant murine M-CSF and TNF-α (from the CMG14-12 cell line) were acquired as previously described [26,54].

### 4.2. Histological Examination

A previous in vivo study suggested that supracalvarial administration of LPS (100 μg) over 5 consecutive days successfully triggered osteoclast formation [55]. We adopted the same protocol for the present study. Mice were randomly allocated to four groups (4 mice/group) and administered supracalvarial injection of PBS, LPS alone (100 μg), LPS (100 μg) with (D-Ala^2^)GIP (25 nmol/kg b.w.), and (D-Ala^2^)GIP alone (25 nmol/kg b.w.) for 5 days. The calvariae of the mice were dissected after euthanization on day 6 and fixed with 4% formaldehyde for 7 days at 4 °C, followed by demineralization in 14% ethylenediaminetetraacetic acid (EDTA) at room temperature for 7 days. After dehydration by a tissue processor (TP1020, Leica, Wetzlar, Germany), the calvariae were embedded into paraffin and sliced by a microtome (Leica) into 5 μm sections perpendicular to the sagittal suture. The sections were first stained with a TRAP staining kit, following the manufacturer’s instructions (FUJIFILM Wako Pure Chemical Corporation, Osaka, Japan), and then counterstained with hematoxylin. TRAP-positive cells with three or more nuclei located on the mesenchyme of the sagittal suture were identified as osteoclasts [55].

### 4.3. Micro-CT Evaluation of Bone Destruction Area

Following euthanasia on day 6, the calvariae of the mice were fixed in 4% formaldehyde immediately after dissection. After 7 days of fixation, micro-computed tomography (micro-CT) (ScanXmate-E090; Comscan, Kanagawa, Japan) and Software TRI/3DBON64 0.9.0.0 (RATOC System Engineering, Tokyo, Japan) were used to scan the calvariae and generate reconstructed images. The black area on the sutures was considered to indicate bone resorption in contrast to the white bone surface. Next, a 50 × 70 pixel rectangle was drawn at the junction of the coronal and sagittal sutures. The proportion of the bone destruction region to the entire surface of the calvaria was evaluated using ImageJ (NIH, Bethesda, MD, USA) (https://imagej.net/ij/, accessed on 4 November 2022) [55].

### 4.4. Osteoclast Preparation

Bone marrow cells were obtained from the femora and tibiae of 8–10-week-old male C57BL6/J mice. The bones were cut at both ends and centrifuged twice at 4 °C in α-Minimum Essential Medium (α-MEM; Sigma-Aldrich). After filtering through a 40 μm cell strainer (Falcon, New York, NY, USA), the collected cells were cultured in M-CSF (100 ng/mL)-added α-MEM, which was complemented with 10% fetal bovine serum (FBS), 100 IU/mL penicillin G (Meiji Seika, Tokyo, Japan), and 100 µg/mL streptomycin (Meiji Seika) for 3 days. Adherent cells were considered bone marrow macrophages, a type of osteoclast precursor, and harvested using trypsin-EDTA solution (Sigma-Aldrich). The obtained cells were plated in four groups as 4 × 10^4^ cells/well in a 96-well plate; each group was incubated with M-CSF only (100 ng/mL), M-CSF (100 ng/mL), and RANKL (100 ng/mL) or TNF-α (100 ng/mL), M-CSF (100 ng/mL), and RANKL (100 ng/mL) or TNF-α (100 ng/mL) with (D-Ala^2^)GIP (200 nM) or M-CSF (100 ng/mL) and (D-Ala^2^)GIP (200 nM) for 5 days. Media were replenished every two days. Cells were fixed with 4% formaldehyde for 30 min, followed by a 30 min permeabilization with 0.2%Triton-X-100. A TRAP solution was prepared by mixing acetate buffer (pH 5.0), naphthol AS-MX phosphate (Sigma-Aldrich), Fast Red Violet LB Salt (Sigma-Aldrich), and 50 mM sodium tartrate and used to stain the cells. Cells with three or more nuclei that were positive for TRAP were identified as osteoclasts.

### 4.5. Cell Viability Assay of Osteoclast Precursors

Osteoclast precursors were seeded evenly as 1 × 10^4^ cells/well in a 96-well plate and divided into four groups. Each group was treated with M-CSF (100 ng/mL) only or with M-CSF (100 ng/mL) and different concentrations of (D-Ala^2^)GIP (2, 20, and 200 nM). After a 5-day incubation period, the CCK-8 solution (Cell Counting Kit-8; Dojin, Kumamoto, Japan) was added. Cells were further cultured for 2 h at 37 °C, and absorbance was measured at 450 nm with a microplate reader.

### 4.6. Preparation of Peritoneal Macrophages

Sterile, cold PBS (6 mL) was injected into the peritoneal cavity of 8–10-week-old C57BL6/J male mice. The same fluid was aspirated to obtain peritoneal macrophages under resting conditions. Following centrifugation twice at 4 °C, cells were filtered through a 40 μm cell strainer. After 2 h of culture, PBS was used to rinse and remove non-adherent cells. After 24 h of culture, the adherent cells were harvested and used as peritoneal macrophages [55].

### 4.7. Preparation of Primary Osteoblasts

Calvariae were resected from 7–9 d new-born C57BL6/J mice. The collagenase solution was prepared in an isolation buffer (3 mM K_2_HPO_4_, 10 mM NaHCO_3,_ 60 mM sorbitol, 70 mM NaCl, 1 mM CaCl_2_, 0.5% [*w*/*v*] glucose, 0.1% [*w*/*v*] bovine serum albumin[BSA] and 25 mM 4-(2-hydroxyethyl)-1-piperazineethanesulfonic acid). 5 mM EDTA was prepared with 0.1% BSA in PBS and strained through a 0.2 µm filter. Then, the calvariae were digested by collagenase solution and EDTA for 20 and 15 min, respectively, on a 37 °C shaker. Fractions 2 (EDTA), 3 (collagenase), 4 (collagenase), and 5 (EDTA) and their digests were considered osteoblast-rich and collected. The obtained cells were cultured overnight, and adherent cells were harvested using trypsin-EDTA on day 2. The medium was changed every 2 days. After incubation for another 3 days, adherent cells were regarded as osteoblasts and used in subsequent experiments [56].

### 4.8. RNA Preparation and Real-Time RT-PCR Evaluation

For in vivo experiments, dissected calvariae were first frozen in liquid nitrogen and then shattered in 800 μL TRIzol reagent (Invitrogen, Carlsbad, CA, USA) via Micro Smash MS-100R (Tomy Seiko, Tokyo, Japan). Next, the RNeasy Mini Kit (Qiagen, Valencia, CA, USA) was used to extract total RNA. For in vitro experiments, peritoneal macrophages or osteoblasts were cultured in four groups, each receiving PBS, LPS alone (100 ng/mL), LPS (100 ng/mL), and (D-Ala^2^)GIP (20 nM), or (D-Ala^2^)GIP (20 nM) alone for 3 days and 1 day, respectively. The same RNeasy Mini Kit was used to isolate total RNA from all cells. cDNA was synthesized using oligo-dT primers (Invitrogen) and reverse transcriptase. The expression levels of TRAP, cathepsin K, TNF-α, and RANKL mRNA were analyzed by RT-PCR using a Thermal Cycler Dice Real-Time System (Takara Bio, Shiga, Japan). The reaction mixture (25 μL) comprised 2 μL of cDNA sample and 23 μL SYBR Premix Ex Taq (Takara, Shiga, Japan) and 50 pmol/μL primers. The primers used in this study were as follows: GAPDH: 5′-GGTGGAGCCAAAAGGGTCA-3′ and 5′-GGGGGCTAAGCAGTTGGT-3′; Cathepsin K: 5′-GCAGAGGTGTGTACTATGA-3′ and 5′-GCAGGCGTTGTTCTTATT-3′; TRAP: 5′-AACTTGCGACCATTGTTA-3′ and 5′-GGGGACCTTTCGTTGATGT-3′; TNF-α: 5′-CTGTAGCCCACGTCGTAGC-3′ and 5′-TTGAGATCCATGCCGTTG-3′; RANKL: 5′-CCTGAGGCCCAGCCATTT-3′ and 5′-CTTGGCCCAGCCTCGAT-3. ’ The expression levels of TRAP, cathepsin K, TNF-α, and RANKL mRNA were standardized to that of GAPDH.

### 4.9. Western Blotting Analysis

To investigate the effects of (D-Ala^2^)GIP on the phosphorylation of p38, ERK1/2, and JNK mitogen-activated protein kinases (MAPKs) in peritoneal macrophages and osteoblasts, Western blotting evaluation was performed. Cells were cultured in 60 mm culture dishes overnight, followed by a 6 h serum-free starvation period. LPS alone (100 ng/mL) or LPS with (D-Ala^2^)GIP (20 nM) was added to the dishes and stimulated for 0, 5, 15, and 30 min. Neither LPS nor (D-Ala^2^)GIP was added to control dishes (0 min). Cells were lysed in radioimmunoprecipitation (RIPA) assay buffer (Millipore, Burlington, MA, USA) containing 1% protease and phosphatase inhibitors (Thermo Fisher Scientific, Waltham, MA, USA) for 20 min, and the insoluble substance was removed instantly by centrifugation. The acquired protein samples were then subject to treatment by a mixture of β-Mercaptoethanol (BioRad, CA, USA) and Laemmli sample buffer (BioRad, CA, USA) at a 3:1 proportion and denatured for 5 min at 95 °C. Next, an equal quantity of proteins was loaded into ten-well Mini-PROTEAN TGX Precast Gels (BioRad, CA, USA), transferred to a 0.2 μm PVDF via Trans-Blot Turbo Transfer System (Bio-Rad, CA, USA), and then incubated for 1–2 h at room temperature in BlockAce (DS Pharma Biomedical, Osaka, Japan). Membranes were incubated with the following antibodies: Phospho-SAPK/JNK (Thr183/Tyr185) rabbit mAb, phospho-p44/42 (ERK1/2) MAPK (Thr202/Tyr204) rabbit mAb, phospho-p38 MAPK (Thr180/Tyr182) rabbit mAb (Cell Signaling Technology, Danvers, MA, USA) (diluted 1:3000), and anti-β-actin mAb (Sigma-Aldrich, MO, USA) (diluted 1:5000) overnight at 4 °C. After being rinsed using tris-buffered saline with Tween-20(TBS-T), the membranes were incubated using anti-rabbit IgG, HRP-linked antibody (Cell Signaling Technology, Danvers, MA, USA) (diluted 1:5000), or ECL peroxidase-labelled anti-mouse antibody (diluted 1:10,000) (Cytiva, Tokyo, Japan) at room temperature for 1 h. Finally, a chemiluminescence-based detection process was conducted using SuperSignalWest Femto Maximum Sensitivity Substrate (Thermo Fisher Scientific, IL, USA).

### 4.10. Statistical Analysis

All data were displayed as average values and standard deviations. The difference between groups was evaluated using the Tukey–Kramer test and paired *t*-test, and statistical significance was set at *p* < 0.05.

## 5. Conclusions

Our results demonstrate that (D-Ala^2^)GIP inhibited LPS-induced inflammatory osteoclast formation and bone resorption in vivo by dampening TNF-α expression in macrophages and RANKL expression in osteoblasts as well as by directly suppressing osteoclastogenesis. These findings provide promising insights into the role of (D-Ala^2^)GIP in osteoclast formation and bone resorption in inflammation-related bone diseases and may contribute to the development of novel therapies for osteolysis in patients with diabetes.

## Figures and Tables

**Figure 1 ijms-25-02555-f001:**
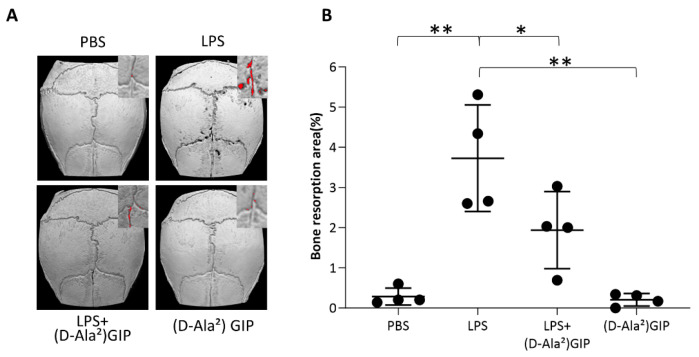
(D-Ala^2^)GIP suppressed LPS-induced bone resorption in vivo. (**A**) Reconstructed mouse calvarial structure pictures using micro-CT after subcutaneous injection of PBS, LPS, LPS+(D-Ala^2^)GIP, or (D-Ala^2^)GIP alone for 5 d. Red spots indicate areas of bone destruction. (**B**) The proportion of resorbed bone region to total calvaria area. Data are presented as mean values ± SD. Tukey–Krammer analysis was adopted to evaluate the significance of variances (*n* = 4; * *p* < 0.05, ** *p* < 0.01).

**Figure 2 ijms-25-02555-f002:**
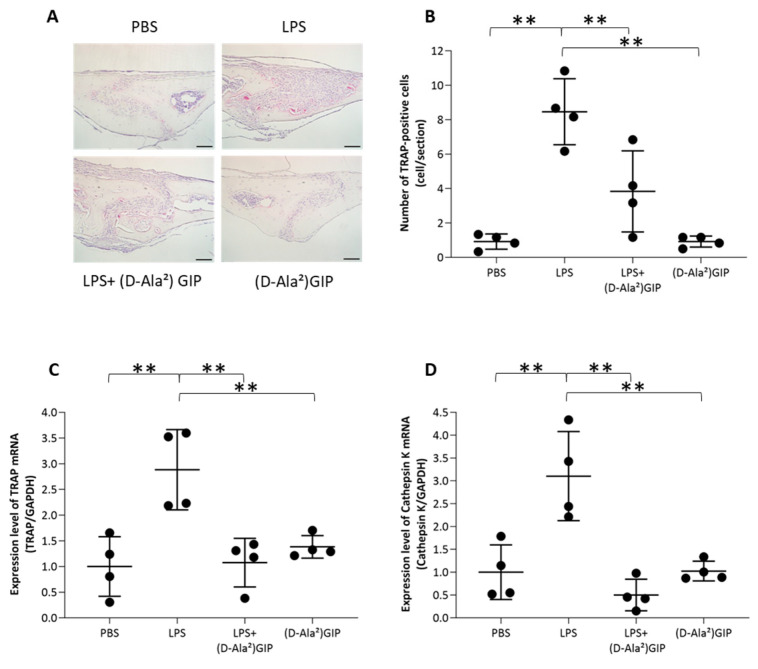
(D-Ala^2^)GIP suppressed LPS-induced osteoclast formation in vivo. (**A**) Mouse calvariae stained by TRAP to identify osteoclasts after subcutaneous administration of PBS, LPS, LPS+(D-Ala^2^)GIP, or (D-Ala^2^)GIP alone for 5 d. After TRAP staining, histological sections were counterstained with hematoxylin. (**B**) Amount of TRAP-positive multinucleated cells in mesenchymal tissue of mouse calvarial sagittal sutures. Scale bar = 50 μm. (**C**) TRAP mRNA expression (relative to GAPDH) in mouse calvariae. (**D**) Cathepsin K mRNA expression (relative to GAPDH) in mouse calvariae. Data are presented as mean values ± SD. Tukey–Krammer analysis was adopted to evaluate the significance of variances (*n* = 4; ** *p* < 0.01).

**Figure 3 ijms-25-02555-f003:**
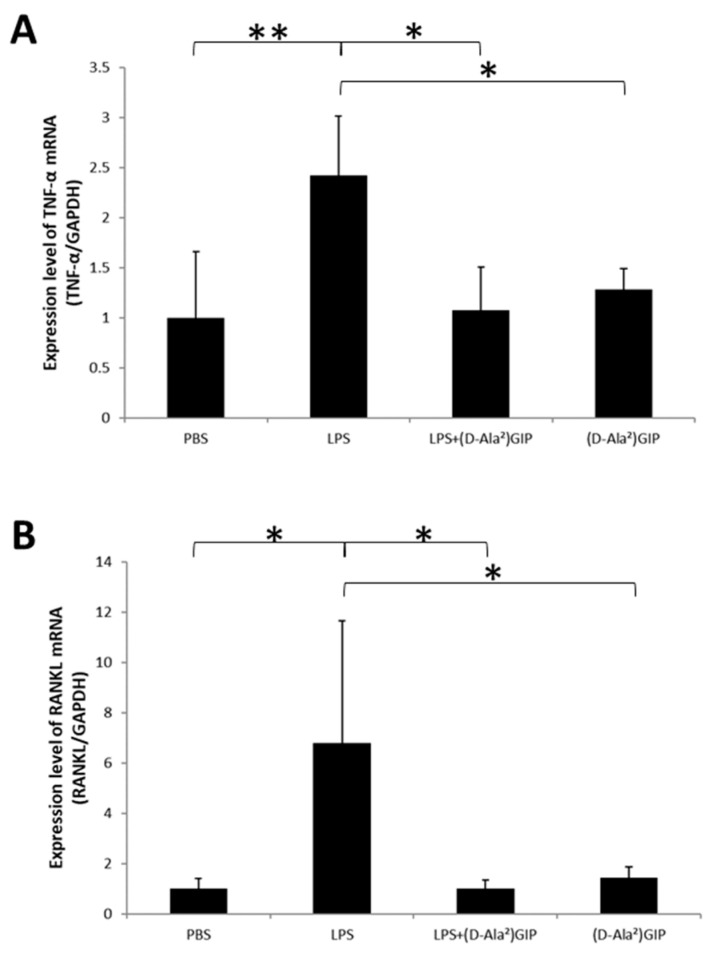
(D-Ala^2^)GIP suppressed LPS-induced mRNA expression of TNF-α and RANKL in vivo. Real-time RT-PCR was conducted to evaluate expression level of the two cytokines in mouse calvariae, following administration of PBS, LPS, LPS+(D-Ala^2^)GIP or (D-Ala^2^)GIP alone for 5 d. (**A**) TNF-α mRNA expression (relative to GAPDH) in mouse calvariae. (**B**) RANKL mRNA expression (relative to GAPDH) in mouse calvariae. Data are presented as mean values ± SD. Tukey–Krammer analysis was adopted to evaluate the significance of variances (*n* = 4; * *p* < 0.05, ** *p* < 0.01).

**Figure 4 ijms-25-02555-f004:**
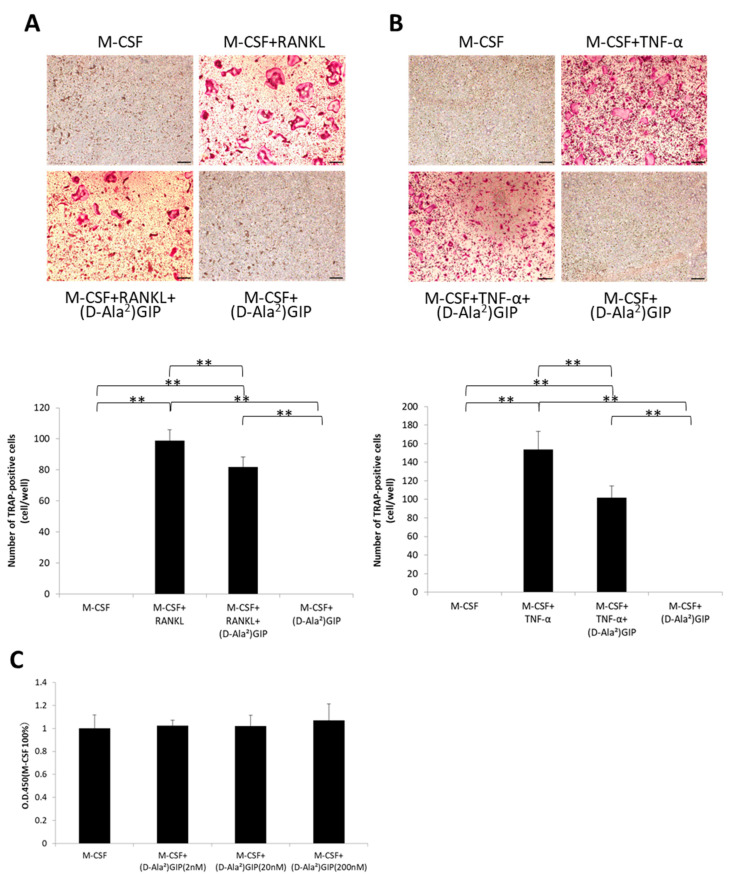
(D-Ala^2^)GIP suppressed osteoclastogenesis triggered by RANKL and TNF-α and had no impact on osteoclast precursor viability in vitro. Bone marrow cells were obtained from the femora and tibiae of the mice and were further cultured with M-CSF for 3 days to become osteoclast precursors. (**A**) Images and number of TRAP-positive cells after being incubated with M-CSF, M-CSF+ RANKL, M-CSF+RANKL+(D-Ala^2^)GIP, or M-CSF+(D-Ala^2^)GIP for 5 d (*n* = 3). Scale bar = 200 μm. (**B**) Images and number of TRAP-positive cells after being incubated with M-CSF, M-CSF+ TNF-α, M-CSF+ TNF-α+(D-Ala^2^)GIP, or M-CSF+(D-Ala^2^)GIP for 5 d (*n* = 4). Scale bar = 200 μm. (**C**) Cell viability of osteoclast precursors cultivated with M-CSF alone or M-CSF with different concentrations of (D-Ala^2^)GIP for 5 d (*n* = 4). Data are presented as mean values ± SD. Tukey–Krammer analysis was adopted to evaluate the significance of variances (** *p* < 0.01).

**Figure 5 ijms-25-02555-f005:**
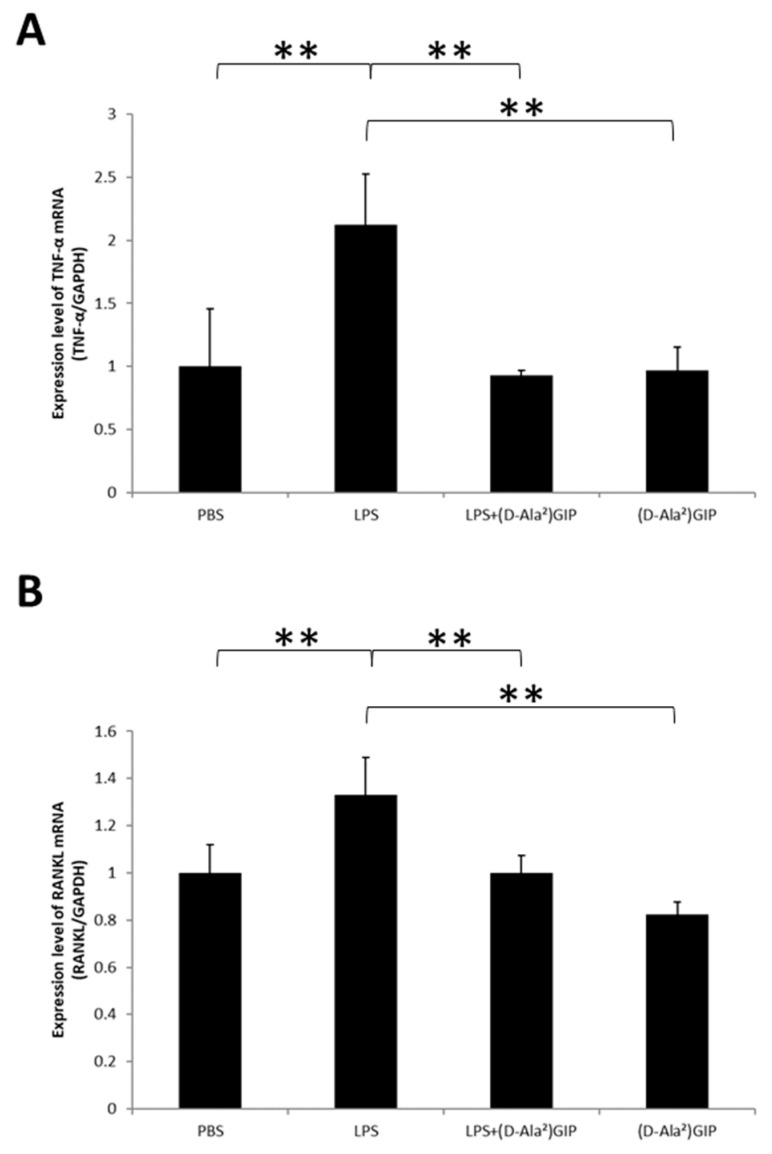
(D-Ala^2^)GIP downregulated TNF-α and RANKL expression triggered by LPS in peritoneal macrophages and osteoblasts, respectively, in vitro. (**A**) TNF-α mRNA expression in peritoneal macrophages incubated with PBS, LPS, LPS+(D-Ala^2^)GIP, or (D-Ala^2^)GIP alone for 3 d. (**B**) RANKL mRNA expression in osteoblasts incubated with PBS, LPS, LPS+(D-Ala^2^)GIP, or (D-Ala^2^)GIP alone for 1 day. Data are presented as mean values ± SD. Tukey–Krammer analysis was adopted to evaluate the significance of variances (*n* = 4; ** *p* < 0.01).

**Figure 6 ijms-25-02555-f006:**
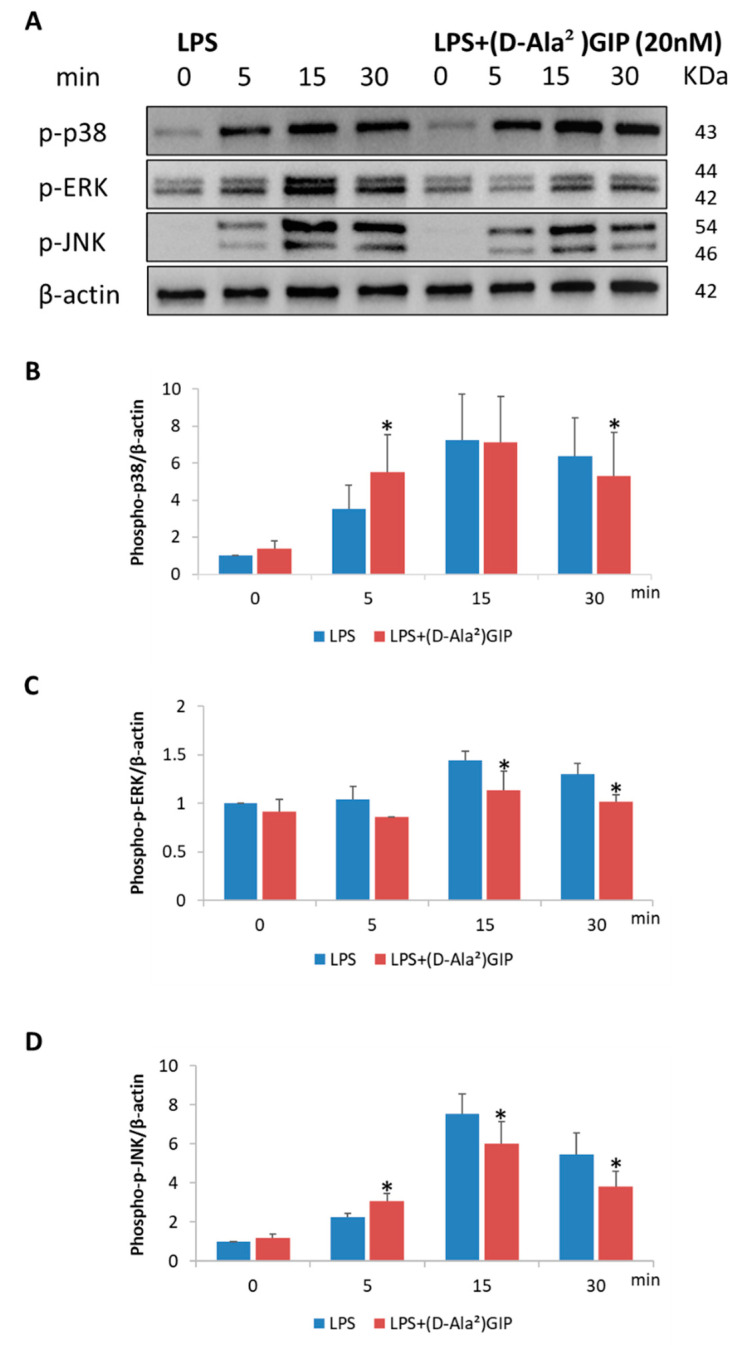
(D-Ala^2^)GIP impaired LPS-induced MAPKs pathway activation in peritoneal macrophages in vitro. Peritoneal macrophages were starved for 6 h and then treated with LPS alone or LPS+(D-Ala^2^)GIP for 0, 5, 15, and 30 min. Proteins were isolated and used in Western blotting to quantify phospho-p38/ERK/JNK and β-actin. (**A**) Band images of phospho-p38/ERK/JNK and β-actin. (**B**) Quantification of phospho-p38 relative to β-actin. (**C**) Quantification of phospho-ERK relative to β-actin. (**D**) Quantification of phospho-JNK relative to β-actin. Data are presented as mean values ± SD. Paired *t*-test was adopted to evaluate the significance of variances (*n* = 3; * *p* < 0.05).

**Figure 7 ijms-25-02555-f007:**
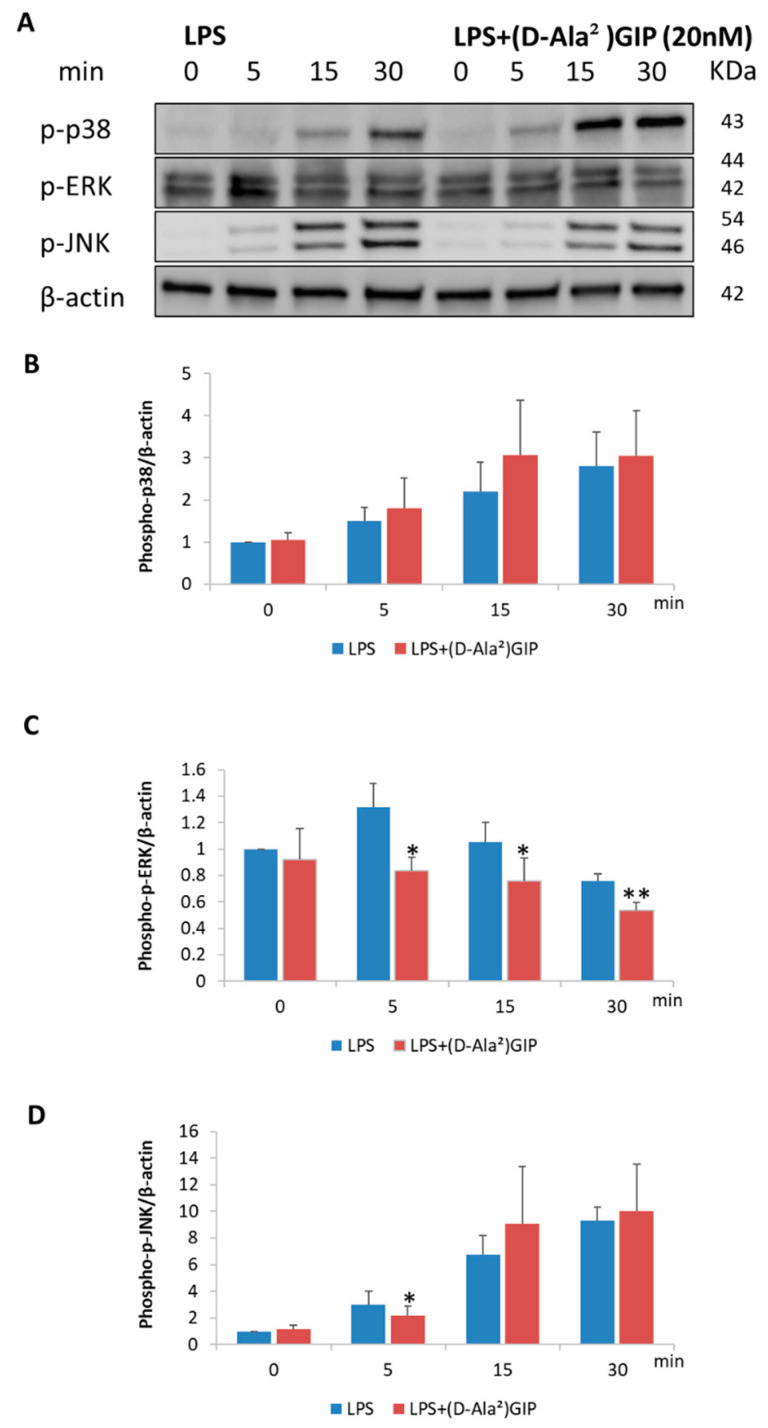
(D-Ala^2^)GIP impaired LPS-induced MAPKs pathway activation in osteoblasts in vitro. Osteoblasts were starved for 6 h and then treated with LPS alone or LPS+(D-Ala^2^)GIP for 0, 5, 15, 30 min. Proteins were isolated and used in Western blotting to quantify phospho-p38/ERK/JNK and β-actin. (**A**) Band images of phospho-p38/ERK/JNK and β-actin. (**B**) Quantification of phospho-p38 relative to β-actin. (**C**) Quantification of phospho-ERK relative to β-actin. (**D**) Quantification of phospho-JNK relative to β-actin. Data are presented as mean values ± SD. Paired *t*-test was adopted to evaluate the significance of variances (*n* = 3; * *p* < 0.05, ** *p* < 0.01).

**Figure 8 ijms-25-02555-f008:**
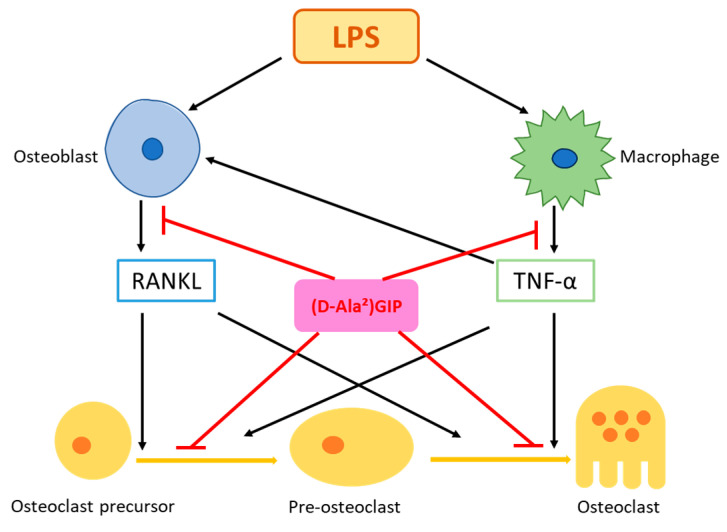
Illustration of potential (D-Ala^2^)GIP mechanism in LPS-triggered osteoclast formation and bone resorption in vivo. LPS enhances the secretion of RANKL from osteoblasts and TNF-α from peritoneal macrophages. The two proinflammatory cytokines are crucial for promoting osteoclastogenesis, promoting the transformation of osteoclast precursors to pre-osteoclasts and then osteoclasts. (D-Ala^2^)GIP prevents LPS-induced production of RANKL in osteoblasts and TNF-α in peritoneal macrophages and exerts a direct inhibitory effect on osteoclast formation, thus mitigating osteoclastogenesis and bone resorption driven by LPS in vivo.

## Data Availability

Data is contained within the article.

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
