# Peer review of "(D-Ala2)GIP Inhibits Inflammatory Bone Resorption by Suppressing TNF-α and RANKL Expression and Directly Impeding Osteoclast Formation"

_ijms, 2024, doi:10.3390/ijms25052555_

Round 1

Reviewer 1 Report

Comments and Suggestions for Authors

Lin et al. evaluated the effect of (D-Ala2)GIP on LPS-induced osteoclast formation and bone resorption in vivo and explored its possible mechanisms via in vitro experiments. The results show that (D-Ala2)GIP significantly suppressed LPS-triggered osteoclastogenesis and bone resorption by modulating the expression levels of RANKL and TNF-α in vivo. Further in vitro study showed that (D-Ala2)GIP directly impeded osteoclastogenesis induced by RANKL and TNF-α without affecting the cell viability of osteoclast precursors. As the potential mechanisms underlying the inhibition of TNF-α and RANKL expression by (D-Ala2)GIP in vivo, the authors suggested that (D-Ala2)GIP might suppressed the MAPKs signaling pathway.

Overall, the aim of study is clear and the presented results seem solid to draw the conclusion which authors state in the abstract.

I believe the manuscript to be accepted for publication in IJMS.

Author Response

Lin et al. evaluated the effect of (D-Ala2)GIP on LPS-induced osteoclast formation and bone resorption in vivo and explored its possible mechanisms via in vitro experiments. The results show that (D-Ala2)GIP significantly suppressed LPS-triggered osteoclastogenesis and bone resorption by modulating the expression levels of RANKL and TNF-α in vivo. Further in vitro study showed that (D-Ala2)GIP directly impeded osteoclastogenesis induced by RANKL and TNF-α without affecting the cell viability of osteoclast precursors. As the potential mechanisms underlying the inhibition of TNF-α and RANKL expression by (D-Ala2)GIP in vivo, the authors suggested that (D-Ala2)GIP might suppressed the MAPKs signaling pathway.

Overall, the aim of study is clear and the presented results seem solid to draw the conclusion which authors state in the abstract.

I believe the manuscript to be accepted for publication in IJMS.

Thank you for reading our article and giving the comment. We sincerely appreciate your recommendation.

Reviewer 2 Report

Comments and Suggestions for Authors

This manuscript by Lin A et al., presented data on role of (D-Ala2)GIP in bone resorption in inflammation. Authors treated mice with (D-Ala2)GIP under LPS induced inflammation. Authors have chosen osteoclast formation and bone resorption as the primary readouts. Additionally, authors evaluated the expression levels of TNF-a, and RANKL. Finally, authors found that (D-Ala2)GIP treatment had protective effect by lowering the bone resorption. Authors concluded that (D-Ala2)GIP effect was mediated by TNF-a and RANKL down regulation, thereby lowered inflammation.

This manuscript is of particular interest. Abstract and the manuscript is nicely written, however, the data presentation is must be improved.

Below are my minor concerns:

1) The primary readouts are crucial to confirm the effect of (D-Ala2)GIP. authors have chosen TRAP IHC and micro-CT to evaluate this. In my opinion, authors could perform at least another method to strengthen their findings. eg., may be osteocalcin or similar staining could be easy performed, since the authors have readily available sections. 

2) I suggest to change the order of the figures, ie. Fig 2 should be placed as Fig 1. Followed by Osteocalcin or similar marker staining, and Fig 3 will be Fig 1 of the present version. This way, the manuscript flow sounds better.

3) Surprisingly, micro-CT representative images are at poor quality and the magnified area images are out of the focus. 

4) Please add time point of at which day the micro-CT images were taken.

5) Please change Fig 1 and Fig 2 graphs to scatter plots. 

Major concern:

1) If I got correct, mice were sacrificed at 6th day after the treatment. It's hard to convince myself that micro-CT and IHC analysis were able to detect the bone deformity within this given short window frame.

2) Diabetes mellitus associated with systemic inflammation, and authors have also pointed bone resorption burden in diabetes mellitus patients. However, authors have chosen LPS-induced inflammation model. What is rationale? using db/db mouse model would have been greatly advantageous over chosen approach. 

Author Response

To Reviewer 2:

Thank you very much for your constructive comments and suggestions. I sincerely appreciate them and will answer them in order.

Minor concerns:

1) The primary readouts are crucial to confirm the effect of (D-Ala2)GIP. authors have chosen TRAP IHC and micro-CT to evaluate this. In my opinion, authors could perform at least another method to strengthen their findings. eg., may be osteocalcin or similar staining could be easy performed, since the authors have readily available sections.

Thank you for your suggestion. Osteocalcin is a marker for osteoblast activity. In the present study, we aimed to focus more on osteoclasts, specifically, osteoclast formation and osteoclastic bone resorption. Therefore, we didn’t conduct osteocalcin staining. We used TRAP staining to analyze osteoclast formation and evaluated bone resorption by micro-CT in vivo. Furthermore, we carried out PCR analysis. We measured the expression level of TRAP and Cathepsin K, which are two important osteoclast markers for assessing bone resorption activity, to strengthen our elucidation.

2) I suggest to change the order of the figures, ie. Fig 2 should be placed as Fig 1. Followed by Osteocalcin or similar marker staining, and Fig 3 will be Fig 1 of the present version. This way, the manuscript flow sounds better.

Thank you for your suggestion. We have exchanged the order of Figure 1 and Figure 2, as well as narration and figure description.

3) Surprisingly, micro-CT representative images are at poor quality and the magnified area images are out of the focus. ,

Thank you for your comment. We have revised the pictures, and they are now at better quality.

4) Please add time point of at which day the micro-CT images were taken.

Thank you for your suggestion. Mice were euthanized on day 6, and the calvariae were fixed by formalin solution for 7 days. Then the micro-CT images were taken. The timepoint is now added to the manuscript(line 325).

5) Please change Fig 1 and Fig 2 graphs to scatter plots.

Thank you for your comment. We have changed the graphs in Figure 1 and Figure 2 to scatter plots.

Major concern:

1) If I got correct, mice were sacrificed at 6th day after the treatment. It's hard to convince myself that micro-CT and IHC analysis were able to detect the bone deformity within this given short window frame.

Thank you for your comment. LPS is a strong inflammation-inducing reagent that can trigger significant osteoclast formation and bone resorption in bone tissues. This is a frequently used method in previous studies, so we adopted it and applied it in mouse calvariae. To confirm the effect of LPS, we used micro-CT to evaluate the bone destruction(Figure 1). In the LPS group, the rough and uneven bone surface as a result of severe bone resorption can be recognized. The short window frame was used to highlight the resorption area in the suture mesenchyme. We also conducted TRAP staining of the calvaria sections(Figure 2). In the LPS group, a broadened gap can be detected in the suture, suggesting the extensive bone breakdown. In addition, we did PCR experiment to assess the expression level of TRAP and Cathepsin K, the commonly used markers for evaluating osteoclast activity(Figure 2). Higher expression level of TRAP and Cathepsin K was detected in the LPS group, compared to the control group, indicating significant bone resorption triggered by LPS.

2) Diabetes mellitus associated with systemic inflammation, and authors have also pointed bone resorption burden in diabetes mellitus patients. However, authors have chosen LPS-induced inflammation model. What is rationale? using db/db mouse model would have been greatly advantageous over chosen approach.

Thank you for your question and suggestion. In diabetes mellitus, bone resorption is elevated. And when treated by anti-diabetes drugs, the bone destruction is recovered. In this study, we aimed to investigate whether (D-Ala2)GIP can ameliorate severe bone resorption. Therefore, LPS was used to establish an inflammation model with significant osteoclast formation and bone degradation, so the influence of (D-Ala2)GIP could accordingly become easier to recognize. Meanwhile, we were also interested in whether (D-Ala2)GIP could contribute to the development of therapies for inflammatory bone diseases, apart from its primary effect in decreasing blood glucose levels. If (D-Ala2)GIP could alleviate inflammation-triggered bone destruction, it might in addition help in treating diabetic patients with accompanying inflammatory bone diseases.

Reviewer 3 Report

Comments and Suggestions for Authors

The study addresses an important area of research by investigating the effects of (D-Ala2)GIP on osteoclast formation and bone resorption, particularly in the context of inflammation-related conditions. The use of a GIP analog that is resistant to DPP-4 degradation represents a new approach to improve the therapeutic potential of GIP in bone diseases.

Although the abstract provides a good overview of the results of the study, it lacks specific information on the experimental models used and the summary of the impact of this study.

Introduction part:

11. A more targeted approach that leads directly to the aim of the study could improve readability and relevance. For example, an early emphasis on the link between diabetes, bone health and the potential of GIP analogs could provide a clearer narrative pathway to the aims of the study.

22. Several previous studies on GIP and its analogs are cited in the introduction, describing effects on both bone resorption and inflammation. However, the extent of detail and the way in which these studies are presented varies, making it difficult to assess their direct relevance to the aim of the study. A more uniform level of detail for each study cited or a focus on the studies most relevant to the research question could improve coherence.

33. Although the introduction refers to conflicting results regarding the effects of GIP on adipose tissue inflammation, it does not elaborate on how these discrepancies might affect the interpretation of the role of GIP in bone tissue inflammation and osteoclast formation.

Results part

4. The authors used different treatments to mice, why a control group receiving no treatment (no PBS) was not included? E.g. to establish baseline bone resorption and osteoclast formation levels

5. Diabetes and chronic inflammation are known to increase the risk of bone fractures and complications. The authors have shown that (D-Ala2)GIP leads to the downregulation of TNF-α and RANKL during LPS-induced acute inflammation. Can you elaborate on how these results might extend to or be representative of chronic inflammation scenarios, particularly in diabetic conditions?

Materials and methods

6. Is there a statistical justification for this sample size to ensure sufficient power to detect significant differences between groups? (4 mice were used)

7. Could the authors provide more details on the randomization process for assigning mice to experimental groups? Were the experiments conducted in a blinded manner to minimize bias in outcome assessment?

8. What concentrations of LPS and (D-Ala2)GIP were used, and how were they determined? Also, could you please specify the volume and route of injections?

Author Response

To Reviewer 3:

Thank you very much for your constructive comments and suggestions. I sincerely appreciate them and will answer them in order.

Introduction part:

  1. A more targeted approach that leads directly to the aim of the study could improve readability and relevance. For example, an early emphasis on the link between diabetes, bone health and the potential of GIP analogs could provide a clearer narrative pathway to the aims of the study.

Thank you for your comment. We have reordered our narration and made modifications according to your suggestion.

  1. Several previous studies on GIP and its analogs are cited in the introduction, describing effects on both bone resorption and inflammation. However, the extent of detail and the way in which these studies are presented varies, making it difficult to assess their direct relevance to the aim of the study. A more uniform level of detail for each study cited or a focus on the studies most relevant to the research question could improve coherence.

Thank you for your comment. We have adjusted the extent of presented details about the referred studies according to your suggestion.

  1. Although the introduction refers to conflicting results regarding the effects of GIP on adipose tissue inflammation, it does not elaborate on how these discrepancies might affect the interpretation of the role of GIP in bone tissue inflammation and osteoclast formation.

Thank you for your comment. We have added our thought on the discrepancies in GIP’s effect on adipose tissue inflammation in the discussion part. We also linked the discrepancies to our own findings. The supplemented part is highlighted in red (line 241-248).

Results part

  1. The authors used different treatments to mice, why a control group receiving no treatment (no PBS) was not included? E.g. to establish baseline bone resorption and osteoclast formation levels

Thank you for your question. PBS is a buffer solution and usually used in experiments as the negative control. In the present study, LPS and (D-Ala2)GIP used for in vivo experiments were both dissolved in PBS solution before injection to eliminate the effect PBS could bring. Therefore, we didn’t include a no-PBS group.

  1. Diabetes and chronic inflammation are known to increase the risk of bone fractures and complications. The authors have shown that (D-Ala2)GIP leads to the downregulation of TNF-α and RANKL during LPS-induced acute inflammation. Can you elaborate on how these results might extend to or be representative of chronic inflammation scenarios, particularly in diabetic conditions?

Thank you for your question. In diabetes mellitus, bone tissue is often suspect to chronic inflammation. The constantly elevated expression level of TNF-α and RANKL leads to more osteoclast formation as well as bone resorption in patients, resulting in an osteoporosis condition. In the present study, (D-Ala2)GIP downregulated TNF-α and RANKL expression, thus recovered acute inflammatory bone destruction induced by LPS. (D-Ala2)GIP might also be able to ameliorate bone resorption and improve bone quality by suppressing the expression of the two cytokines in chronic inflammation, if applied in a long-term manner. Further research is required for confirmation. Meanwhile, (D-Ala2)GIP might also exert a beneficial effect in improving bone health in diabetes patients with inflammatory bone diseases, such as rheumatoid arthritis. We have also modified our discussion part according to your suggestion. The modified part is highlighted in red(line 281-289).

Materials and methods

  1. Is there a statistical justification for this sample size to ensure sufficient power to detect significant differences between groups? (4 mice were used)

Thank you for your question. We ensured enough power for significance detection. For in vivo experiments(4 mice were used in each group), we first conducted one-way Anova analysis to compare the differences of means of each group. We made sure that he null hypothesis of equal group means is rejected. Then we used Tukey-Krammer analysis, one of the commonly used methods for multiple comparison test to detect variances between groups, and to specify between which groups the variances occurred. The minimum sample size required for Tukey-Krammer test is n=3. The statistical analyses we carried out are thereby considered sufficient for the adopted sample size.

  1. Could the authors provide more details on the randomization process for assigning mice to experimental groups? Were the experiments conducted in a blinded manner to minimize bias in outcome assessment?

Thank you for your questions. Mice were purchased and all were transported to the animal facility in one box. We randomly pick the mice four by four and assigned them to four same cages. Mice were kept at the animal facility in suitable conditions till they adapted to the environment. Then the cages were labelled respectively PBS, LPS, LPS+(D-Ala2)GIP and (D-Ala2)GIP on a random basis, and the mice in each cage received corresponding reagent treatment. After collecting the samples in each group, we conducted subsequent analysis in a blinded manner. The label of each group was kept as key by another lab member, and the answer was told to the analyzer after the evaluation process was finished. In this way, we minimized possible bias in outcome assessment.

  1. What concentrations of LPS and (D-Ala2)GIP were used, and how were they determined? Also, could you please specify the volume and route of injections?

Thank you for your question. LPS was dissolved in PBS and applied at a concentration of 1 μg/μL. (D-Ala2)GIP was also dissolved in PBS and was applied at a concentration of 0.025 μg/μL. The total volume of injection was 100 μL for each mouse. The reagents were injected subcutaneously near the anterior fontanelle to observe osteoclast formation and bone resorption in the calvaria. The method was established in our lab, and we followed the route of our previous works (Shen et al. J. Immunol. Res., 2018; Ishida et al. Biomed. Pharmacother., 2019; Kishikawa et al. Front. Endocrinol., 2019).

Round 2

Reviewer 2 Report

Comments and Suggestions for Authors

The technical evaluation of hypotheses is still limited. The authors have appropriately responded to all my concerns.